# MBNL1 alternative splicing isoforms play opposing roles in cancer

Tommaso Tabaglio[1,2] , Diana HP Low[1,3], Winnie Koon Lay Teo[1], Pierre Alexis Goy[1,2], Piotr Cywoniuk[4], Heike Wollmann[1], Jessica Ho[1] , Damien Tan[1], Joey Aw[1], Andrea Pavesi[1], Krzysztof Sobczak[4], Dave Keng Boon Wee[5,8], Ernesto Guccione[1,2,3,6,7,]

The extent of and the oncogenic role played by alternative splicing (AS) in cancer are well documented. Nonetheless, only few studies have attempted to dissect individual gene function at an isoform level. Here, we focus on the AS of splicing factors during prostate cancer progression, as these factors are known to undergo extensive AS and have the potential to affect hundreds of downstream genes. We identified exon 7 (ex7) in the *MBNL1* (Muscleblind-like 1) transcript as being the most differentially included exon in cancer, both in cell lines and in patients' samples. In contrast, *MBNL1* overall expression was down-regulated, consistently with its described role as a tumor suppressor. This observation holds true in the majority of cancer types analyzed. We first identified components associated to the U2 splicing complex (SF3B1, SF3A1, and PHF5A) as required for efficient ex7 inclusion and we confirmed that this exon is fundamental for MBNL1 protein homodimerization. We next used splice-switching antisense oligonucleotides (AONs) or siRNAs to compare the effect of *MBNL1* splicing isoform switching with knockdown. We report that whereas the absence of MBNL1 is tolerated in cancer cells, the expression of isoforms lacking ex7 (*MBNL1 Δex7*) induces DNA damage and inhibits cell viability and migration, acting as dominant negative proteins. Our data demonstrate the importance of studying gene function at the level of alternative spliced isoforms and support our conclusion that MBNL1 Δex7 proteins are antisurvival factors with a defined tumor suppressive role that cancer cells tend to down-regulate in favor of *MBNL +ex7* isoforms.

## Introduction

In humans and all other eukaryotes, there is a clear discrepancy between the estimated number of proteins (>100,000; Savage [2015]) and the relatively limited number of genes (~20,300; Genome Reference Consortium [2014]). Alternative splicing (AS) is the process that contributes to this diversity by rearranging coding or noncoding sequences in a highly coordinated and complex fashion (Kornblihtt et al, 2013). What was initially thought to be a regulatory tool involved in the expression of few mammalian genes has been estimated to be an extensively exploited mechanism occurring in ~95% of multi-exonic genes (Pan et al, 2008). De facto, each gene in the human transcriptome has an average of seven alternatively spliced isoforms, whereas this number decreases in lower eukaryotes (*Drosophila melanogaster*: 1.9, *Caenorhabditis elegans*: 1.5) (Lee & Rio, 2015).

AS, therefore, enables eukaryotic cells to increase their potential in gene expression control, cellular differentiation, and signal transduction. As such, misregulation of AS has been linked to specific disease etiology (Kornblihtt et al, 2013).

Nonetheless, this complexity is often overlooked in favor of using bulk transcript abundance when ascribing gene function in different pathological or physiological settings. To address this, we reanalyzed publically available and in-house–generated datasets with a specific focus on the abundances of individual exons as opposed to overall transcript levels. We decided to focus on splicing factors in our analysis, as these proteins are known to exist in multiple splice isoforms, with each isoform having the potential to differentially regulate or interact with hundreds of downstream genes. By doing so, we discovered that whereas *MBNL1* levels are overall down-regulated between normal and cancer tissues, *MBNL1* exon 7 (ex7) inclusion increases in almost all tumor samples.

MBNL1 is a well-studied RNA-binding protein (RBP) involved in splicing, RNA export, and stability (Goers et al, 2010; Tran et al, 2011; Masuda et al, 2012; Konieczny et al, 2014; Sznajder et al, 2016). Whereas its role in cellular differentiation and in the mechanism underlying myotonic dystrophy has been deeply investigated in the past decades (Lee & Cooper, 2009; Timchenko, 2013), its function in cancer has been explored only recently (Fish et al, 2016; Singh et al, 2018). To systematically assess isoforms' function in an endogenous setting, we took advantage of the splice-switching antisense

[1]Institute of Molecular and Cell Biology, Agency for Science, Technology and Research, Singapore   [2]Department of Biochemistry, Yong Loo Lin School of Medicine, National University of Singapore, Singapore   [3]Cancer Science Institute, Singapore   [4]Department of Gene Expression, Institute of Molecular Biology and Biotechnology, Adam Mickiewicz University, Poznan, Poland   [5]Institute of High Performance Computing, Agency for Science, Technology and Research, Singapore   [6]National Cancer Centre Singapore, Singapore   [7]Department of Oncological Sciences, Tisch Cancer Institute, Icahn School of Medicine at Mount Sinai, New York, NY, USA   [8]Bioinformatics Institute, Agency for Science, Technology and Research, Singapore

Correspondence: eguccione@imcb.a-star.edu.sg; ernesto.guccione@mssm.edu

oligonucleotide (AON) technology. These AONs are fully modified RNA-based molecules that do not trigger any enzymatic reaction and do not recruit RNaseH activity, but rather bind to RNA through Watson–Crick base pairing, interfering with RBPs and skewing the splicing reaction in the desired direction.

The general aims of our study were to determine the phenotypical implications of the presence/absence of *MBNL1* ex7 in cancer, while understanding its upstream regulators and downstream molecular mechanisms of action.

# Results

### MBNL1 ex7 is highly included in cancer cells and tissues

We decided to investigate whether the AS of splicing factor genes was changing in cancer tissues. In fact, the AS of splicing factors is an often-overlooked phenomenon that can dramatically influence multiple downstream mRNA targets, in the way they are spliced, their overall abundance, or their cellular localization (Änkö et al, 2012; Lareau & Brenner, 2015). A better understanding on how the differential splicing patterns of splicing factors in cancer can sustain the disease is needed. We analyzed prostate cancer TCGA datasets (PRAD, The Cancer Genome Atlas) looking for differential AS of a panel of 93 splicing factors and RBPs (Fig 1A). These RBPs belong either to the core spliceosome machinery, are considered AS factors, or are simply known to bind RNA and be involved in its metabolism. We calculated the percentage of spliced-in (PSI or $\Psi$) values of every exon of genes in the list and computed the $\Delta$PSI ($\Delta\Psi$) values as $\Psi_{tumor} - \Psi_{normal}$ using the SpliceSeq database (Ryan et al, 2012). The analysis was performed taking into account the average PSI of normal versus tumor, without taking into consideration matching patients' samples. The total list of differential splicing events is shown in Fig 1A and Table S1. Ex7 of the *MBNL1* transcript displayed the largest $\Delta$PSI. This is a very short (36 bp) protein-coding exon with a reported role in homodimerization in yeast (Tran et al, 2011). Comparatively, the overall abundance of *MBNL1* transcripts is only slightly decreased in cancer tissues, in line with what has been described in the literature (Sebestyén et al, 2016) (Fig 1B). In addition, *MBNL1* itself does not appear to be considerably mutated, amplified, or deleted in the prostate cancer datasets analyzed (Fig S1A). Extending our analysis to matched patients' tissues from the TCGA PRAD dataset, *MBNL1* ex7 was found to be consistently more included in cancer compared to normal prostate tissue (Fig 1C), whereas the decrease in overall mRNA transcripts is consistent for most of the patients (Fig 1D).

We then proceeded to check whether this splicing event is differentially included in two prostate cancer cell lines (LnCap and PC3) compared to untransformed epithelial prostate cells (PrEC). Paired-end RNA-sequencing (RNA-seq) libraries from all the described cell lines confirmed the higher inclusion of ex7 in cancer cells, as compared to normal (Fig 1E, red boxes). The RNA-seq results were validated by individual capillary fragment-length analysis of RT–PCR (FLA-PCR) (Fig 1E, black boxes), a method that allows to distinguish exon inclusion/skipping, based on the entire transcript length, at a single base resolution. Overall, these cell lines recapitulate the *MBNL1* ex7 inclusion pattern seen in cancer patients, making them a good model to study the molecular basis of this splicing event.

Following these observations, we expanded our analysis to assess the level of *MBNL1* ex7 inclusion in other cancer types. We used the SpliceSeq data to score for AS alterations of the aforementioned splicing factors' panel in multiple TCGA datasets, besides PRAD. Strikingly, *MBNL1* ex7 was among the highest included exons between normal and cancer tissues in all analyzed datasets (Fig 1F). The first conclusion is that this exon might represent an important molecular event to initiate and/or sustain tumor growth across multiple cancer types.

### MBNL1 ex7 is enhanced by U2 overexpression

*MBNL1* is characterized by multiple, complex AS isoforms (Fig S1B). To understand what upstream regulatory factors are responsible for including ex7 in *MBNL1*, we performed a targeted siRNA screen in PC3 cells. We used pooled siRNAs targeting a panel of 64 splicing factors and RBPs (including controls) (Fig 2A). We measured how the individual knockdown of these 64 RBPs affected the inclusion of ex7 using a TaqMan-based assay. Our experimental strategy enabled us to simultaneously quantify the presence of transcripts including ex7 (MBNL1 +ex7) and excluding ex7 (MBNL1 $\Delta$ex7). A probe for the quantification of the housekeeping gene GAPDH was included, allowing for analysis and normalization of multiple samples in parallel. Our results indicate that various members of the U2 snRNP are involved in direct modulation of *MBNL1* ex7 inclusion. Specifically, *SF3B1* and *SF3A1*, together with *PHF5A* (followed by *PRPF8*), topped the list of genes that, when depleted, promote *MBNL1* ex7 skipping (>threefold increase in $\Delta$ex7/+ex7 ratio) (Fig 2A, green bars). An increase in *MBNL1* $\Delta$ex7 isoforms and the decrease in ex7-containing isoforms occurred upon depletion of these genes, but did not significantly alter overall MBNL1 abundance (Fig S2A). Notably, SF3B1, SF3A1, and PHF5A all interact with each other and belong to the main U2 spliceosome complex (Finci et al, 2018). Interestingly, U2AF65/U2AF2, an additional member of the U2 complex, is also ranking high in this screen, whereas U2AF35/U2AF1 has an opposite effect. Consistent with what is known in the literature (Konieczny et al, 2018), decrease in the *MBNL1* transcript (MBNL1 siRNA) leads to increased ex7 inclusion (Fig 2A, red bar), while SRSF1 also ranks high in the screen as a potential competitor of *MBNL1* autoregulation (Cywoniuk et al, 2017).

We next decided to validate our findings using two separate shRNAs directed against each of the top three targets of the siRNA screening (*SF3B1*, *SF3A1*, and *PHF5A*). We used shRNAs targeting *hnRNPR* and *PTBP1* as negative controls. Given the general role of U2 in both constitutive splicing and AS, we calculated the PSI for all *MBNL1* exons using FLA-PCR on samples transduced with individual shRNA. We observed a significant decrease in the inclusion of ex7 upon depletion of SF3B1, SF3A1, and, to a minor extent, PHF5A, confirming the results from the siRNA screen (Figs 2B and S2B and D). To further validate the involvement of U2 components as responsible of *MBNL1* ex7 inclusion, we used meayamycin, a compound that targets the U2 component SF3B (Albert et al, 2009; Gao et al, 2013). PC3 cells treated with meayamycin at 10 nM concentration for 20 h display strong *MBNL1* ex7 skipping (Fig S2C).

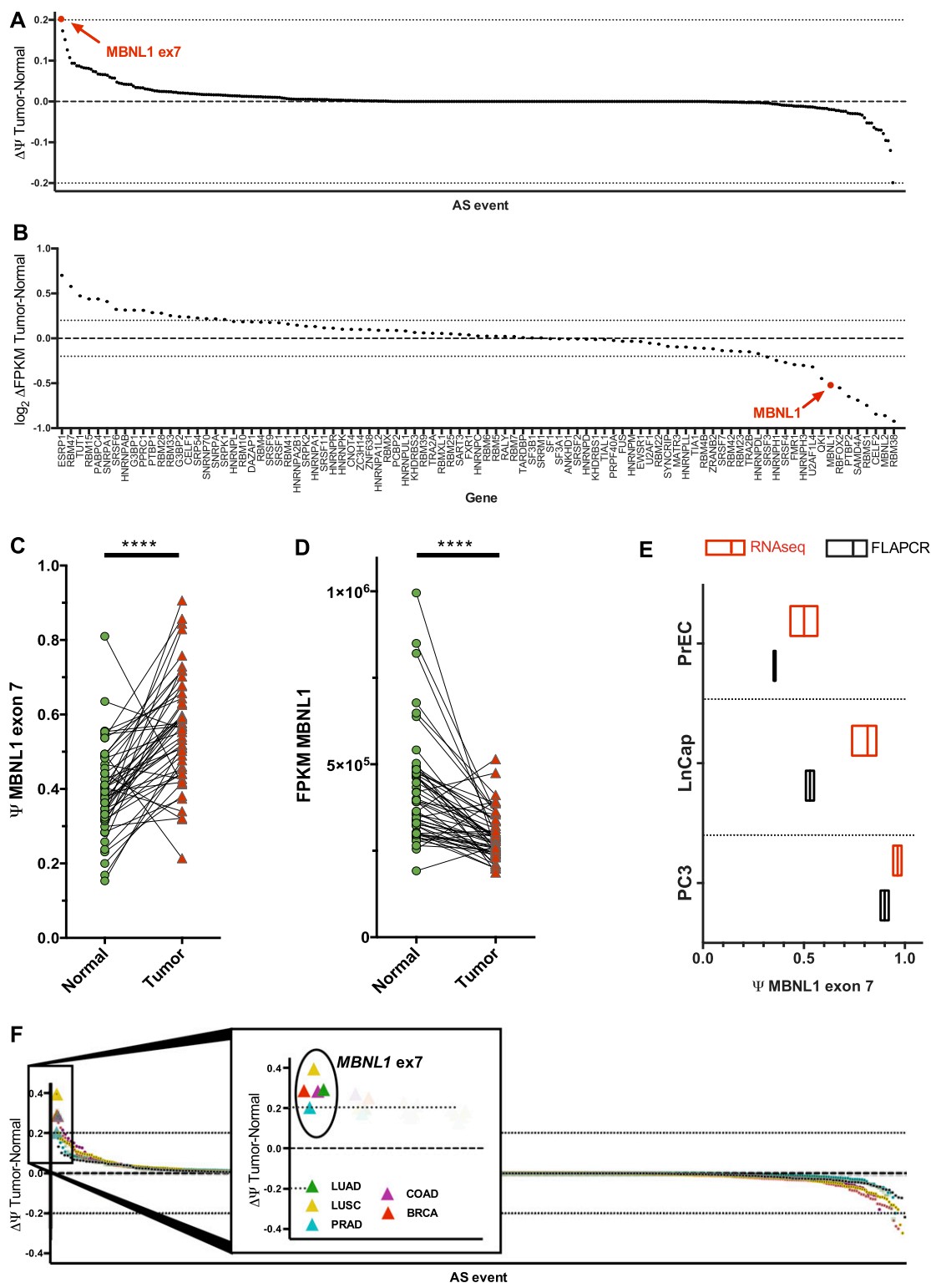

**Figure 1.  *MBNL1* exon 7 is highly included in the mRNA cancer samples and cell lines.**
**(A)** ΔPSI (ΔΨ) of splicing factors' splicing events in the TCGA prostate cancer dataset (PRAD) and **(B)** their relative gene expression change between normal and tumor samples. **(C)** PSI of *MBNL1* ex7 in matched tissues from the TCGA PRCA dataset. **(D)** Overall expression abundance of *MBNL1* in matched tissues from the TCGA PRCA dataset. Statistics performed with two-tailed, parametric, paired *t* test. **(E)** *MBNL1* ex7 PSI values in normal prostate (PrEC) and prostate cancer cell lines obtained through RNA-seq (red) or using FLA-PCR data (black). Data of the FLA-PCR are representative of two independent experiments. **(F)** dPSI of splicing factors' splicing events in various TCGA datasets. *MBNL1* ex7 event enlarged in the central graph for the different datasets. BRCA, breast carcinoma; COAD, colon adenocarcinoma; LUAD, lung adenocarcinoma; LUSC, lung squamous cell carcinoma; PRAD, prostate adenocarcinoma.

**A**

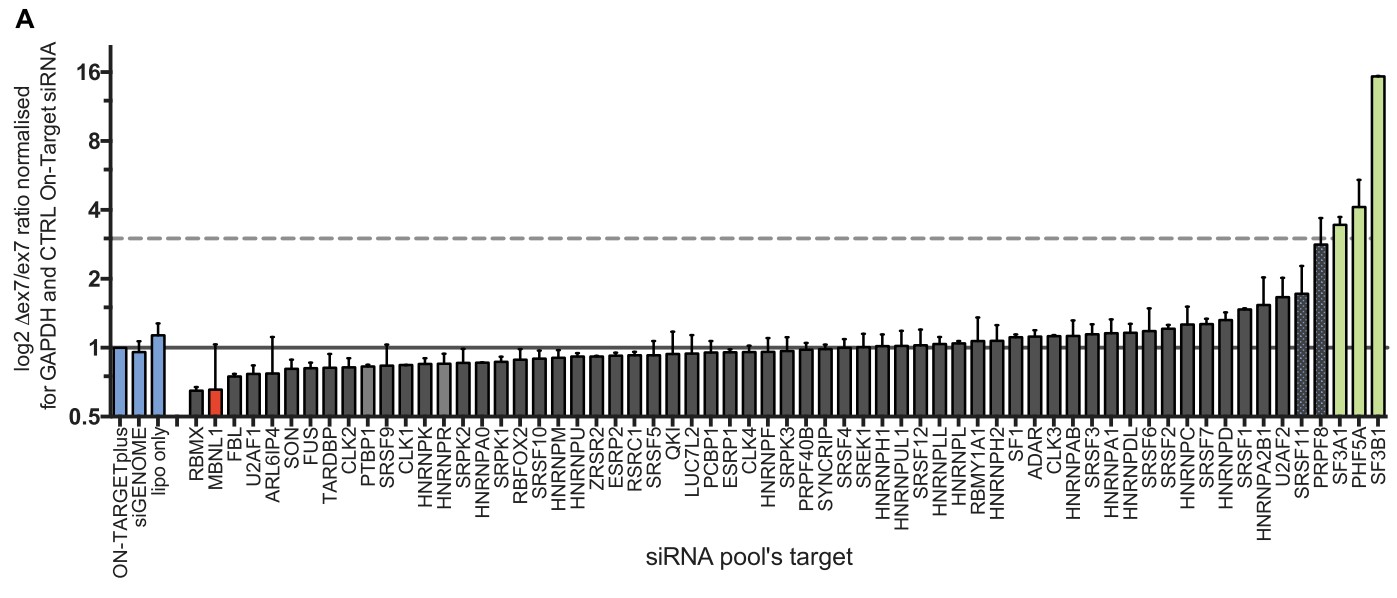

**B**

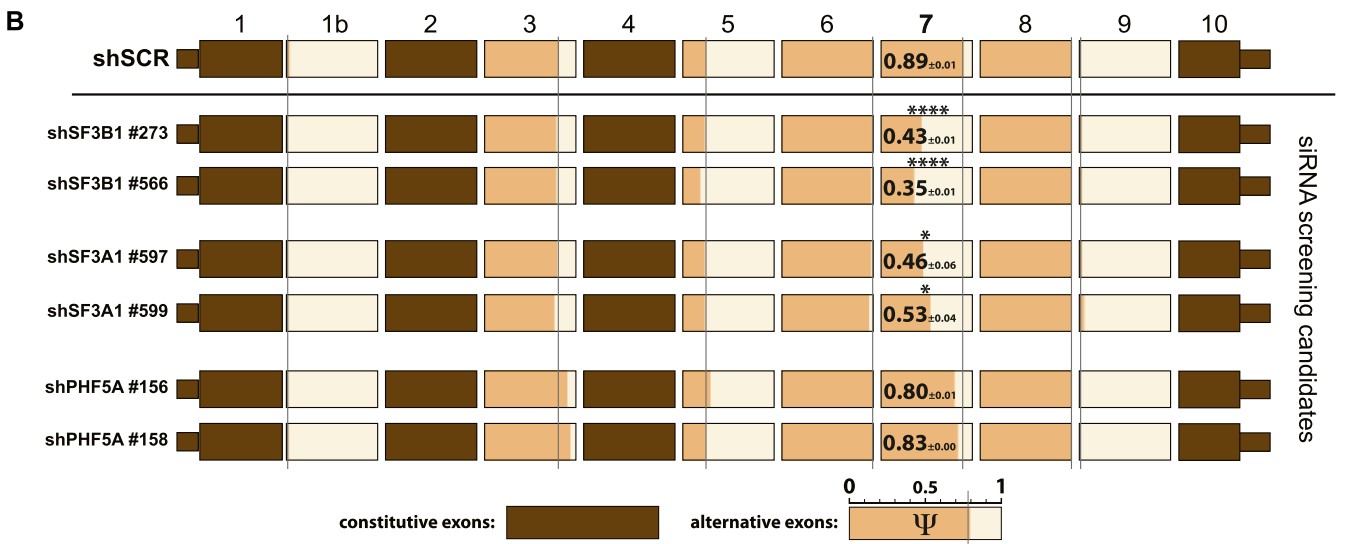

**C**

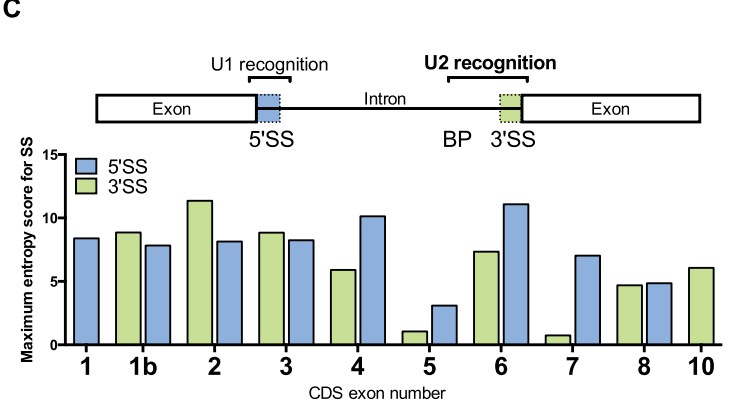

**D**

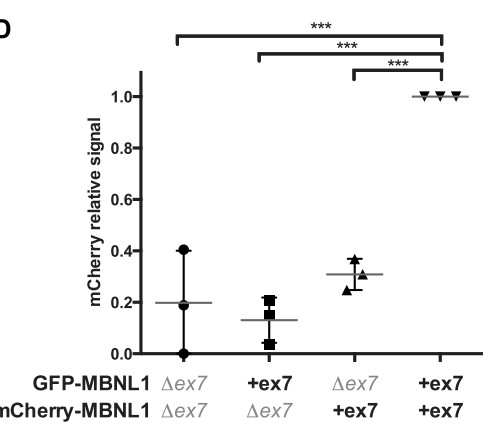

Considering that ex7 inclusion is affected when components of the U2 splicing complex are decreased or inhibited, we evaluated the strength of the 3′SS (recognized by U2) and 5′SS consensus using MaxEntScan (Yeo & Burge, 2004). As expected, *MBNL1* ex7 has the lowest MaxEnt score for the 3′SS, corresponding to a weak 3′SS (Fig 2C), suggesting a reduced likelihood to be recognized by U2 and subsequently included in the mature mRNA. *MBNL1* exon 5 has a comparable low score for the 3′SS, and it is almost completely skipped in the cell lines analyzed. Consistently, we did not observe a further significant change in its exclusion upon knockdowns of U2 components or treatment with meayamycin (Figs 2B and S2C), possibly also because of the involvement of other RBPs in ex5, but not ex7, inclusion.

### MBNL1 ex7 is essential for homodimerization

To understand whether ex7-coded peptide is involved in multimerization of MBNL1 proteins in mammalian cells, as previously suggested in yeast two-hybrid assays (Yuan et al, 2007; Tran et al, 2011), we performed co-immunopurification of MBNL1 proteins. We transfected MBNL1-coding constructs with MBNL1 fused to eGFP or mCherry tags differing in the presence/absence of an ex7-encoded region (isoforms C and H, Fig S1B) in COS7 cells.

These experiments demonstrated that the sequence encoded by ex7 significantly intensifies interactions between MBNL1 proteins and that this process is not RNA dependent (high RNase A treatment of the samples) (Fig 2D). Indeed, dimers of MBNL1 containing ex7 remained stable even in semi-denaturing conditions, where samples were not denatured by heating at high temperature before loading on a gel but were solved in traditional SDS–PAGE. With this method, stable dimers/multimers formed in cells are preserved (Fig S2E and Table S2).

### The use of AONs to selectively induce endogenous MBNL1 ex7 skipping

Chemical inhibition (meayamycin) or shRNA/siRNA-mediated down-regulation of some of the components of the U2 spliceosome complex leads to the increased expression of *MBNL1* isoforms deprived of ex7 (Δex7). However, these approaches have the severe limitation of affecting multiple downstream targets at once.

To selectively perturb *MBNL1* ex7 inclusion, we rationally designed three different steric hindrance AONs, all with a phosphorothioate backbone and 2′-*O*-methyl ribose modifications in each position. These fully modified AONs bind to the target pre-mRNA sequence in the nucleus to mask splicing regulatory cis-elements (Koh et al, 2015; Dewaele et al, 2016). The consequence is an AS "switch" from one splicing isoform to another, without direct degradation of the target

transcript. To test the efficiency of the splice switch of the three AONs upon transfection with lipofectamine-based reagents, we performed a FLA-PCR on the cDNA of transfected PC3 cells. As a control, we used a scrambled, nontargeting AON sequence (Fig 3A). The FLA-PCR enables the quantification of the abundance of every possible splicing isoform, allowing to precisely define the PSI of every exon (Fig 3B, FLA-PCR on AON scrambled [AON SCR] and AON #592). All three AONs reduced ex7 inclusion, with AON #592 giving the best ΔPSI (~90%). It must be considered that the other MBNL family members share high sequence similarity with MBNL1 ex7. In fact, both MBNL2 (ex7) and MBNL3 (ex6) have a highly conserved, alternatively spliced 36-bp exon (Fig S3A). We verified the inclusion of these exons upon AON transfection and proved that AON #592 (from now on called AON MBNL1) is not only giving the best MBNL1 ex7 skipping, but also leaves MBNL2 ex7 and MBNL3 exon 6 mostly unaltered (Fig S3B). We repeated the same experiment in LnCap cells and consistently observed that whereas the isoforms containing ex7 decreased upon AON MBNL1 transfection, the isoforms without ex7 showed a proportional increase (Fig S3C). The AON-induced isoform switch is reflected at the protein level, with isoforms C and G, predominantly present in PC3, being converted to isoforms H and N (Figs 3C and S1B). The mainly nuclear cellular localization of MBNL1, in accordance with previous reports (Tran et al, 2011), does not change in the presence or absence of ex7 (Fig S3D).

### MBNL1 isoforms lacking ex7 act as dominant negative tumor suppressors

AON MBNL1–treated PC3 cells show a marked decrease in cell viability (Fig 3D). Given that AONs can show different degrees of toxicity per se, we used both untransfected and scramble AON–treated cells as controls (Fig 3D, blue circles). Transfecting PC3 cells with the three different AONs directed against ex7 of MBNL1, we observed that cell lethality is proportional to the efficiency of ex7 skipping compared with scramble AON–treated cells (Fig 3E). This effect is dose-dependent and spans across all the cell lines tested (Fig S3E and F). Multiple studies have highlighted the importance of exons 3 and 5 (ex5) for the function and localization of MBNL1, respectively (Tran et al, 2011; Edge et al, 2013). Whereas exon 3 is highly included in all the cell lines tested (PSI > 0.8), ex5 inclusion can fluctuate. To assess whether the phenotype we observed is due to changes in ex5 inclusion upon transfection with AON directed against ex7, we verified that ex5 splicing was not affected by our ex7-skipping AONs in LnCap and PC3 cells. In addition, we designed an AON that efficiently induces the skipping of ex5 and scored for cell viability. Our results clearly demonstrate that our AONs directed against ex5 or ex7 are highly specific (Fig S3G) and that ex7 skipping is correlated with significant reduction in cell viability in

**Figure 2. *MBNL1* exon 7 requires abundant U2 spliceosome components to be included in the mature transcripts and it is necessary for the homodimerization of the protein.**
**(A)** siRNA screening results on *MBNL1* ex7 splicing. Ratio of *MBNL1* ex7-excluding (Δex7) over ex7-containing (+ex7) isoforms normalized for GAPDH and siRNA control upon siRNA directed against a panel of splicing factors. Data represented are obtained from two biological replicates. **(B)** shRNA validation of *MBNL1* ex7 modulator's candidates from siRNA screening in PC3 cells. The PSI of every alternative exon is shown in orange. Constitutive exons are colored in brown. For ex7, the values of the PSI are written in black. Three biological replicates for each shRNA are shown. Paired *t* test. All nonstarred exons are not statistically significant. **(C)** Maximum entropy score for 5′SS and 3′SS for all the CDS exons of *MBNL1* gene calculated with MaxEntScan. A higher score corresponds to a higher chance for the splice site to be recognized by the respective spliceosome subunit. **(D)** Co-immunoprecipitation with anti-GFP beads of GFP-MBNL1 and mCherry-MBNL1. Represented in the graph is the band intensity of three independent biological replicates, blotted with an anti-mCherry antibody. The co-overexpression combination of the isoforms C (with ex7) and H (lacking ex7) is shown below the graph. Unpaired *t* test.

**A**

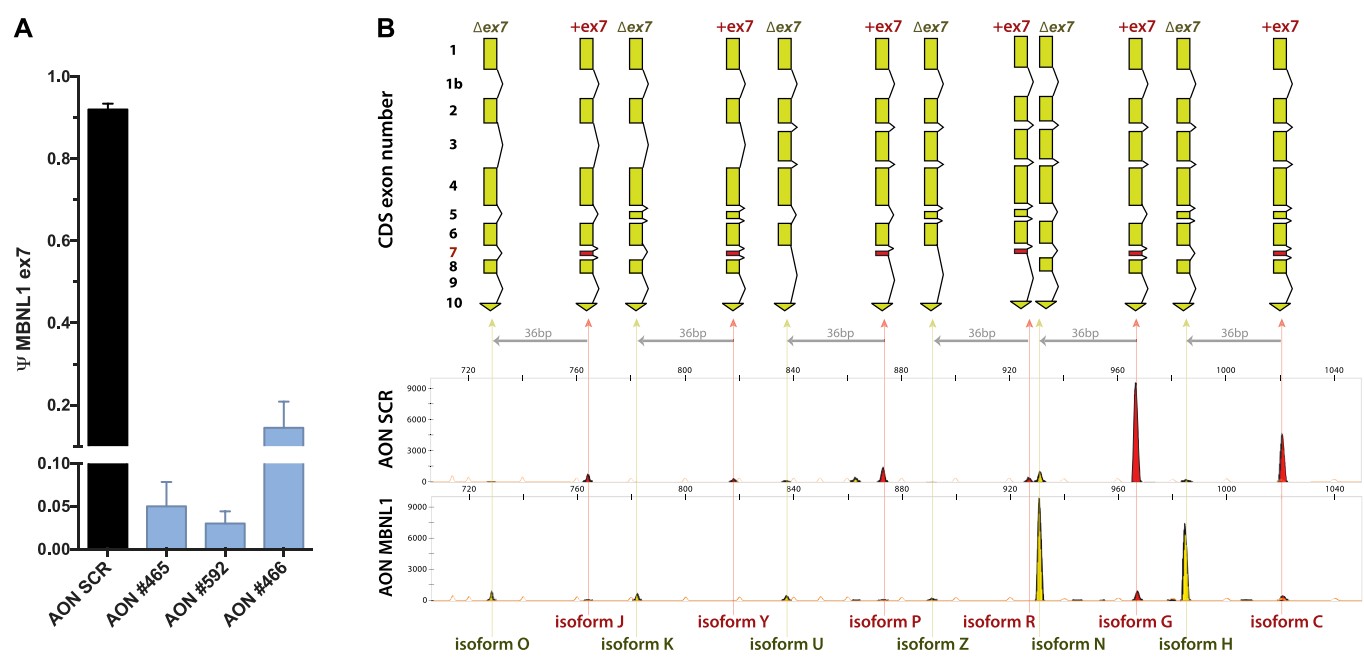

**B**

**C**

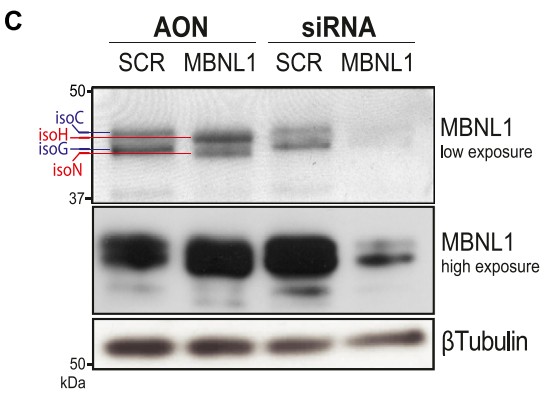

**D**

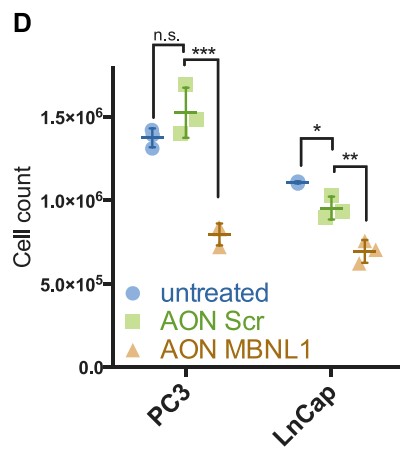

**E**

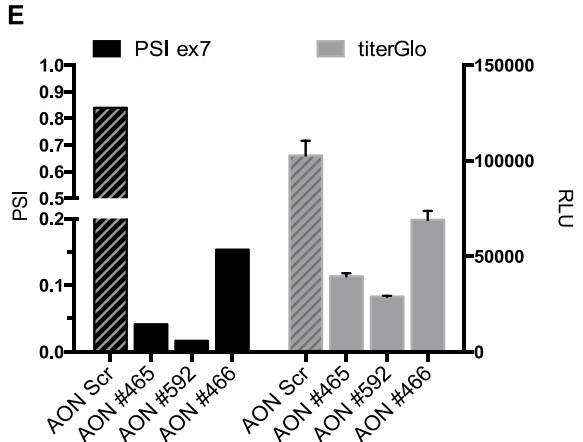

**F**

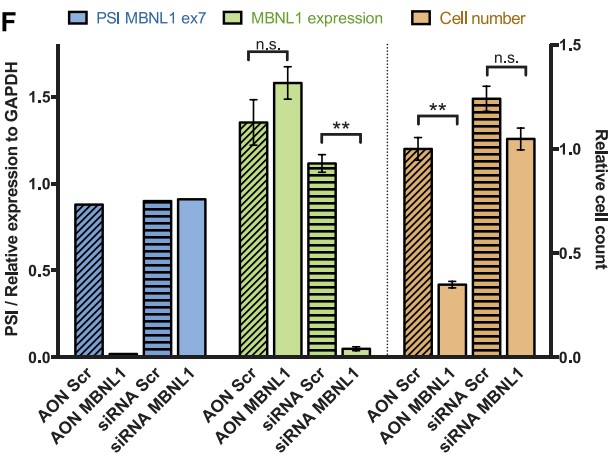

the prostate cancer cells tested, whereas ex5 skipping did not significantly affect cell survival (Fig S3H). To rule out the possibility that the lack of ex7 merely phenocopies the absence of MBNL1 through a general loss of functionality of the protein, we next compared the reduction in cell viability induced by AON MBNL1 with that of pooled siRNA-mediated knockdown. The pooled siRNAs, targeting constitutive exons of the MBNL1 transcript (Fig S3K), reduced the overall protein (Figs 3C and S3I) and transcript levels of *MBNL1* by at least 85% (Figs 3F and S3J). The reduction in cell viability is only observed in the presence of AON MBNL1, whereas MBNL1 siRNA had no observable effect in the two cell lines tested (Figs 3F and S3J).

Importantly, it is possible to rescue the AON-induced lethality with a 6-h pretreatment with siRNA (Fig 4A). We additionally generated a CRISPR-Cas9 MBNL1 knockout PC3 cell line (triploid for MBNL1 gene, Figs S4A and B), which did not show any growth defect upon treatment with AON MBNL1 (Fig 4B). We could confirm in this system that viability upon AON transfection is not significantly compromised in the absence of MBNL1 (Fig 4B). These experiments in addition prove that the AON against MBNL1 is highly specific and that the phenotype observed is not due to off-target effects. These data also suggest that MBNL1 Δex7 is a dominant negative, rather than a loss-of-function, isoform.

### MBNL1 isoforms lacking ex7 regulate specific splicing and transcript abundance

To gain an unbiased mechanistic understanding of the role played by the dominant negative MBNL1 Δex7, we performed an RNA-seq on PC3 cells treated with AON SCR and AON MBNL1 (at 72 h post-transfection) (Fig S5A) and then analyzed the differentially expressed and spliced transcripts. The genes that are up- or down-regulated (*P*- and q-value <0.05, >1 fragments per kilobase of transcript per million mapped reads for at least one of the samples, fold-change >±2) are 127 and 108, respectively (Table S3). The Gene Ontology (GO) analysis (Fig 5A and Tables S4–S6) classifies these genes as involved in the regulation of cell migration, negative regulation of cell proliferation and apoptosis (up-regulated genes), and cell cycle checkpoints and chromosome segregation (down-regulated genes).

AS events, occurring upon treatment with AON MBNL1 compared with control (AON SCR), can be classified as skipped exons (73%), mutually exclusive exons (10%), intron retentions (8%), and alternative 5′-AS5′ (5%) and 3′-AS3′ (4%) splice site selection. These ratios were similar to that of previously reported experiments (Han et al, 2013). In the skipped exons category, 205 exons are more included, whereas 345 are increasingly skipped upon AON MBNL1 treatment compared with control (FDR and *P*-value < 0.05, ΔPSI > 0.1 cutoff) (Table S7).

Genes undergoing AS upon perturbation of *MBNL1* isoform expression fall into GO categories related to actin filament organization, cell division, and DNA repair (Fig 5C).

Following this analysis, we validated several genes belonging to these GO categories by RT–qPCR (Fig 5B and D). Within the selected GO, we validated transcripts that have been demonstrated to be directly bound by MBNL1 based on CLIP-seq experiments (Sznajder et al, 2016). In parallel, samples from siRNA *MBNL1*–transfected PC3 were analyzed to determine whether the event is only due to ex7 inclusion or, more broadly, due to the lack of total MBNL1. In all cases, lack of *MBNL1* (siRNA) did not show any change in the expression of the selected target genes.

### MBNL1 isoforms lacking ex7 decrease cell migration

Given that the top GO categories for alternatively spliced and up-regulated genes were actin filament organization and regulation of cell migration, respectively, we investigated changes in migration potential upon AON MBNL1 transfection. We used a versatile 3D device (Aim Biotech) in which H2B-GFP–labelled, AON-transfected PC3 cells were embedded in a collagen matrix. This system allowed us to monitor (1) the distance covered by the cells during migration and (2) the speed of migration in real time over 14 h. We observed a significant decrease in both the parameters upon treatment with AON MBNL1 compared with AON SCR (track length in AON MBNL1 cells: −16.3%, speed of AON MBNL1 cells: −52.3%) (Figs 6A and S5B). These results point toward the same direction previously seen in another system: Overexpression of the MBNL1 Δex7 isoform (isoform H) reduces dramatically the metastasis size in mouse models (Fish et al, 2016).

### MBNL1 isoforms lacking ex7 induce replication-dependent DNA damage

Besides cell migration and actin reorganization, the most striking GO categories that were significantly enriched in the RNA-seq were related to the regulation of DNA replication and cell cycle: DNA repair, cell division, regulation of chromosome organization (for AS genes), negative regulation of cell proliferation (for up-regulated transcripts), cell cycle, homologous DNA pairing and strand exchange, cell cycle checkpoints, and chromosome segregation (for the down-regulated genes). We, thus, proceeded to investigate whether the cell lethality phenotype is due to increased replicative stress during the S-phase and impaired cell cycle progression. Asynchronous PC3-transfected cells stained with propidium iodide (PI) showed a decreased cell population in G1 (2n) for the AON MBNL1–treated cells, whereas we observed a higher stalling in the S-phase (Fig 6B). Interestingly, the portion of polyploid cells (>4n) is doubled in the AON MBNL1–treated compared with the AON SCR–treated cells (Table S8). These cell cycle/chromosome segregation defects could be linked to increased DNA damage and decreased capability to resolve it during the DNA replication phase (S-phase). To test this hypothesis, transfected cells were stained by immunofluorescence with markers of DNA damage.

**Figure 3. The exclusion of exon 7 through the use of Antisense oligonucleotides reduces cell proliferation.**
**(A)** *MBNL1* ex7 PSI change upon AON transfections (100 nM), 48 h posttransfection. Data are representative of two independent experiments. **(B)** FLA-PCR visualization of *MBNL1* cDNA isoforms upon AON #592 transfection in PC3 cells (100 nM, at 48 h posttransfection). On the X axis is represented the nucleotide size and on the Y axis the fluorescence intensity of the amplicon. Data are representative of multiple independent experiments. **(C)** Western blot of PC3 cells transfected with AON (100 nM) and siRNA (25 nM) directed against MBNL1. Cells were collected 48 h posttransfection. **(D)** PC3 and LnCap absolute cell number at 48 h posttransfection with AON SCR, MBNL1 ex7 (100 nM), or not transfected (no liposome-based reagent, no AON). Statistics performed with two-tailed, parametric, paired *t* test. **(E)** *MBNL1* ex7 PSI change and relative cell viability in PC3 cells. 100 nM of AON after 48 h. Cell viability quantified with TiterGlo luminescence assay. **(F)** Ex7 PSI (blue bars), overall *MBNL1* relative mRNA expression (green bars) and relative cell count of PC3 cells transfected with AON SCR, AON MBNL1 (100 nM), siRNA Scrambled, and siRNA MBNL1 (25 nM).

**A**

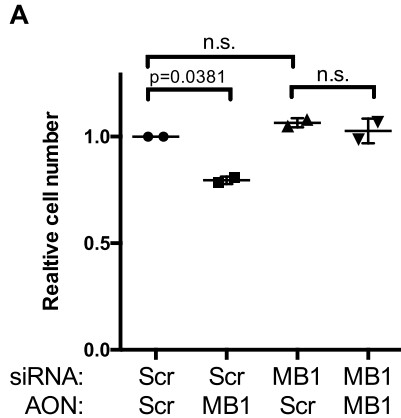

**B**

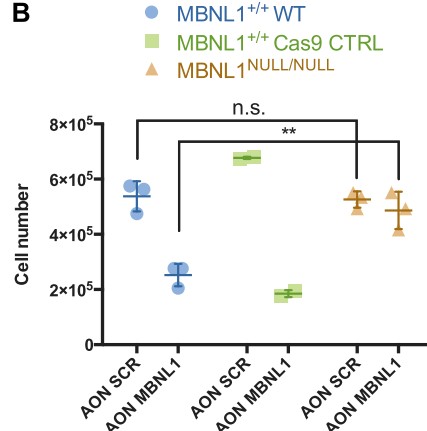

**Figure 4. MBNL1 exon 7-skipping phenotype can be rescued with the depletion of all MBNL1 isoforms. (A)** Relative cell number change upon co-transfection of siRNA against *MBNL1* (50 nM) and ex7-skipping AON (50 nM) in PC3 cells. Parametric paired *t* test. Each dot represents an independent biological replicate. **(B)** Cell count of PC3 cells (blue circles), PC3 cells transfected with Cas9 alone (green squares), and PC3 cells transfected with Cas9 and a sgRNA directed against an MBNL1 coding sequence (orange triangles). The cells were transfected with 75 nM of Scrambled AON and MBNL1 AON and counted 48 h posttransfection. Parametric paired *t* test.

Indeed, cells treated with AON MBNL1 showed a higher number of γH2AX, RAD51, and 53BP1 foci per nucleus, confirming our hypothesis (Fig 6C, D, and E). To investigate whether the DNA replication and damage repair defects are the cause of the decrease in cell viability upon AON MBNL1 treatment, PC3 cells were arrested in G1 with the Cdk4/6 inhibitor PD0332991 (Sigma-Aldrich, 5 μM, in the absence of FBS) for 8 h, then transfected with AON SCR and AON MBNL1. The results demonstrate an almost complete rescue of the cell death for cells arrested in G1 (Fig 6F), corroborating the hypothesis that MBNL1 Δex7 isoforms alter the splicing and the abundance of essential genes for DNA replication, chromosome segregation, and DNA damage repair. Interestingly, some of the genes validated individually are following the same trend (up- or down-regulated/ differentially spliced upon decreased ex7 inclusion) in patients' samples (Fig S5C).

## Discussion

The complexity of AS events in mammalian genomes is becoming increasingly obvious because of the advancement of deep-sequencing technologies and the refinement of computational methods that allow the analysis of such complex datasets. What we believe are now missing are the appropriate tools to functionally test the relevance of such AS isoforms in healthy and disease states.

In this article, we have tried to advance our understanding of disease-relevant isoforms by focusing on the expression of individual exons, rather than cumulative transcript abundance, and reanalyzing publicly available TGCA databases.

We also highlight how splice-switching AON technology can be used to study isoform function in an endogenous setting, avoiding overexpression artifacts. Finally, we adapt capillary electrophoresis (FLA-PCR) to bridge the gap between Illumina sequencing, which, because of the short reads output, can only give information on local exon inclusion levels and a true understanding of exon composition at an isoform level.

Our approach led us to focus on the role of ex7 inclusion in the *MBNL1* transcripts, but can now be extended to other MBNL family members and more in general to any other transcript.

Using a siRNA screening platform against alternative and constitutive splicing factors as well as chemical spliceosome inhibitors, we showed that some of the main components of the U2 spliceosome complex play a fundamental role in *MBNL1* ex7 inclusion. These factors are responsible for the branchpoint and 3′SS recognition. Further analysis of the 3′SS suggests that *MBNL1 ex7* bears the lowest maximum entropy score of the whole transcript, making it an extremely weak site to be recognized by the U2 complex. These findings point at the fact that an abundant and efficient U2 spliceosome complex is required for ex7 inclusion, in accordance with our previous findings demonstrating the importance of high U2 levels in lymphomagenesis (Koh et al, 2015).

In addition, our screen highlights the following: (1) the relevance of SRSF1 as a potential competitor of MBNL1 not only at common downstream targets (Cywoniuk et al, 2017), but also on *MBNL1* autoregulation (low levels of *MBNL1* induce inclusion of ex5 and ex7) and (2) the potential antagonistic role of U2AF65/U2AF2 and U2AF35/U2AF1 (Warf et al, 2009; Echeverria & Cooper, 2014; Cywoniuk et al, 2017) in regulating *MBNL1* ex7 inclusion and possibly other *MBNL1* downstream targets. These observations deserve further investigations.

The use of AON technology allows to study the role of specific isoforms at endogenous levels, while keeping the overall amount of transcript and protein constant and avoiding the overexpression of truncated or specific isoforms (Konieczny et al, 2014; Fish et al, 2016; Sznajder et al, 2016). The AONs designed showed remarkable specificity and efficacy in splice switching.

Notably, neither the AON chemistry itself nor the transfection procedure dramatically affected cell viability. Our data suggest that cell death was instead due to specific ex7 skipping because the use of three different AONs targeting the same exon—with different efficiencies—caused a decrease in the number of viable cells that is proportional to the PSI of ex7. In addition, we can exclude the possibility that the general dysregulation of *MBNL1* splicing influences cell growth and viability because transfection with an AON that excludes exon 5 did not show a relevant phenotype in the two cell lines tested. Besides, it is unlikely that the absence of ex7 simply mimics the absence of the protein, as cells remain viable and are able to proliferate upon MBNL1 siRNA transfection.

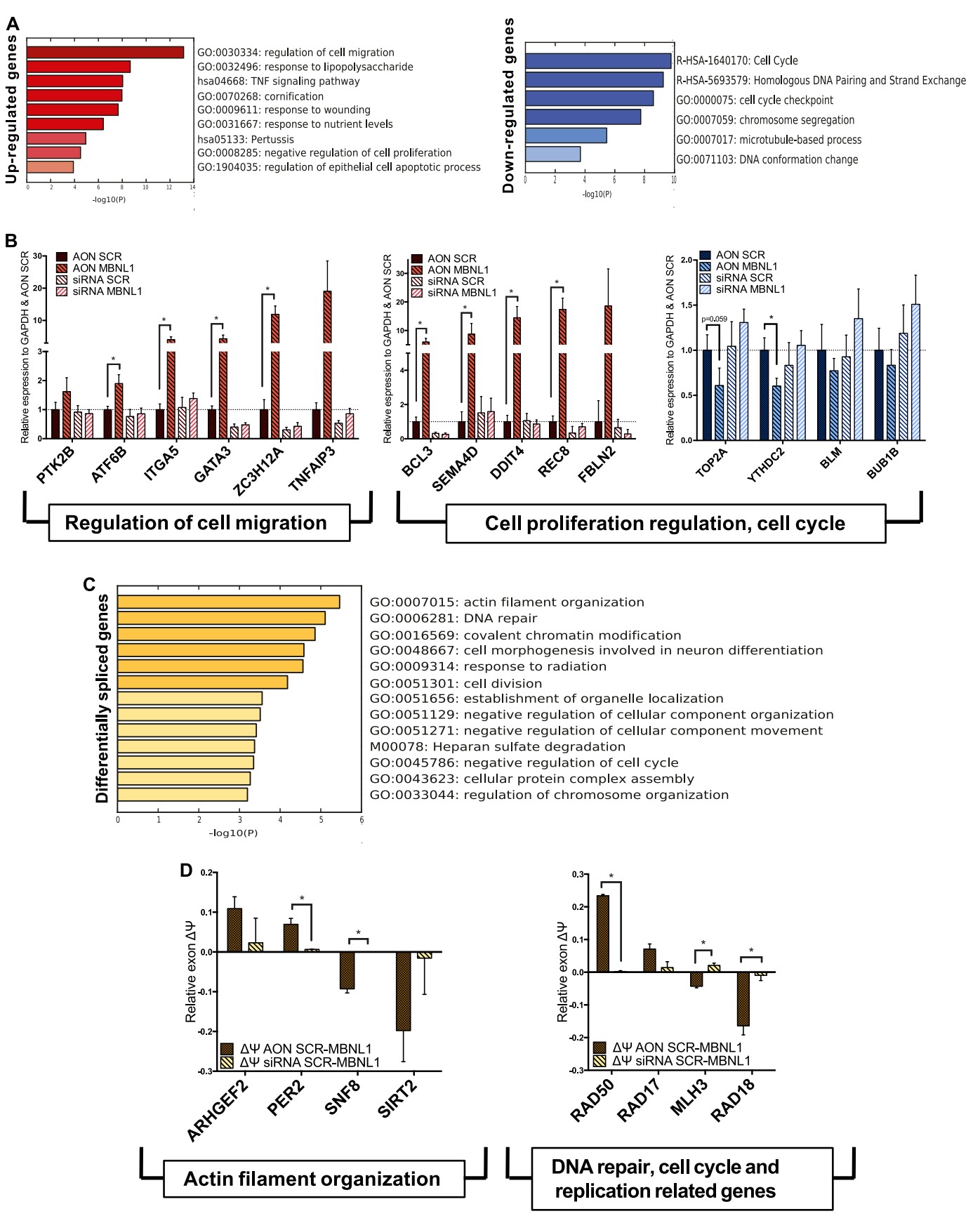

In the interest of understanding the mechanism that links *MBNL1* ex7 with the cell viability phenotype, we performed RNA-seq on PC3 cells treated with AON SCR and AON MBNL1. Remarkably, we could observe a substantial number of splicing events and up- or down-regulated genes involved in mitotic processes, chromosome segregation, DNA replication and repair, cell cycle, cell migration, and actin reorganization. Importantly, although we could validate the results by quantitative PCR using AONs, we did not see the same effect using siRNA to down-regulate all *MBNL1* isoforms at once. Our results strongly highlight the difference between knockdown and selective isoform switching.

In conclusion, we showed that *MBNL1* +ex7 proteins act as dominant negative isoforms, being able to change the abundance and splicing of key genes involved in cell migration, DNA repair, and cell division. Future work has to be carried out to address more in detail the molecular mechanisms underlying this and other important splicing events, both in cancer and in nontransformed cells.

# Materials and Methods

## Cell culture

HEK293T, LnCap, and PC3 were obtained from ATCC and were propagated according to ATCC datasheets. LnCap-CR were generated by culturing LnCap cells in RPMI medium supplemented with 10% charcoal-stripped FBS (#S181F; Biowest) for more than 2 months. PrEC cells were purchased from Lonza (CC-2555) and propagated according to the manufacturer's data sheet.

## Vectors, transfections, and infections

pLKO-1 Mission lentiviral vectors (TRC; Sigma-Aldrich) were used for SF3A1 (TRCN0000006597 and TRCN000000659), SF3B1 (TRCN0000350273 and TRCN0000320566), PHF5A (TRCN0000286156 and TRCN0000286158), hnRNPR (TRCN0000235506 and TRCN0000235508), and PRBP1 (TRCN0000231420 and TRCN0000231421) knock-down in human cell lines. A scrambled shRNA (Scr) was used as a control. HEK293T cells were transfected with pLKO vector together with packaging vectors. The cells were incubated for 18 h and then fresh medium was added to the cells. After 24 h, the medium containing the viral particles was collected, filtered using a 0.22-$\mu$m filter unit, and added onto the target cells. 24 h after the last infection, the medium was replaced with fresh growth medium containing puromycin (540411, Calbiochem; Merck Millipore). Cells were selected 2 d before harvesting.

## AON design

AONs were designed with a rational approach. Briefly, over 40,000 local secondary structures of a target exon and its flanking introns were used to identify RNA sites that are co-transcriptionally accessible for AON binding (Wee et al, 2008; Pramono et al, 2012). The target site of AON #465 was predicted to contain putative splicing enhancer motifs (Fairbrother et al, 2002). AONs #592 and #466 were designed to block the 5′ and 3′ exon splice sites, respectively.

## AON and siRNA transfections

Cells were seeded in 6-well plates at the concentration of 200,000 cells/well. On the following day, the media was replaced with antibiotic-free medium and transfection with AONs (final concentration in the well: 50, 75, or 100 nM according to the experiment) or siRNA (25 or 50 nM according to the experiment, Dharmacon On-Targetplus MBNL1: L-014136-00-0005) was performed using 3 (LnCap) to 4 $\mu$l (PC3, PrEC) of Lipofectamine RNAiMAX transfection reagent (#13778030; Thermo Fisher Scientific) following the manufacturer's instructions. On the subsequent day, the medium was substituted with normal medium containing antibiotics and the cells were collected and counted 48 h posttransfection, if not otherwise stated.

## CRISPR-KO PC3 cell generation

PC3 cells were transfected with the plasmid pSpCas9(BB)-2A-GFP (pX458) (Ran et al, 2013) with the sgRNA shown in Fig S5A. Single cells were expanded and screened for out-of-frame mutation on all the three MBNL1 alleles.

## FACS analysis

Cells were trypsinized and washed with PBS. The collected cells were fixed in 70% ice-cold ethanol overnight at −20 °C. Fixed cells were subsequently stained with 20 $\mu$g/ml PI, 0.1% Triton X-100/PBS, and 0.2 mg/ml RNase A (R6513; Sigma-Aldrich) for 30 min at 22 °C. The cells were then incubated for 30 min at room temperature in 2 M hydrochloric acid/0.1% Triton X-100 to denature the DNA, followed by neutralization with 0.1 M sodium tetraborate, pH 9.0, followed by PI staining and FACS analysis (Becton & Dickinson LSRII flow cytometry analyser).

## Immunofluorescence

Cells were seeded onto poly-D-lysine–coated glass slides. After overnight incubation at 37°C, cells were fixed with 4% PFA/PBS for 15 min at room temperature, permeabilized with 0.25% Triton X-100/PBS, blocked with 1% BSA in 0.1% Triton X-100/PBS, and incubated with primary antibody against the various proteins (Table S9) overnight at 4°C. Secondary antibody incubation was carried out for 1 h at room temperature using secondary AlexaFluor-conjugated antibodies. The slides were mounted with Vectashield mounting

**Figure 5. MBNL1Δ ex7 isoforms affect splicing and stability of several transcripts.**
**(A)** Gene Ontology (GO) of the genes up-/down-regulated (red/blue, respectively) upon AON MBNL1 transfection in PC3 cells. **(B)** RNA-seq validation by qPCR for the selected subset of up-/down-regulated transcripts. In the left panel are grouped the genes involved in cell migration, whereas in the right panel are shown the genes involved in DNA replication and repair, and chromosome segregation. Data show the results of three independent biological replicates. **(C)** Gene Ontology of the genes alternatively spliced (yellow) upon AON MBNL1 transfection in PC3 cells. **(D)** RNA-seq validation by qPCR for the selected subset of differentially spliced transcripts. In the left panel are grouped the genes involved in cell migration, whereas in the right panel are shown the genes involved in DNA replication and repair, and chromosome segregation. Data show the results of three independent biological replicates.

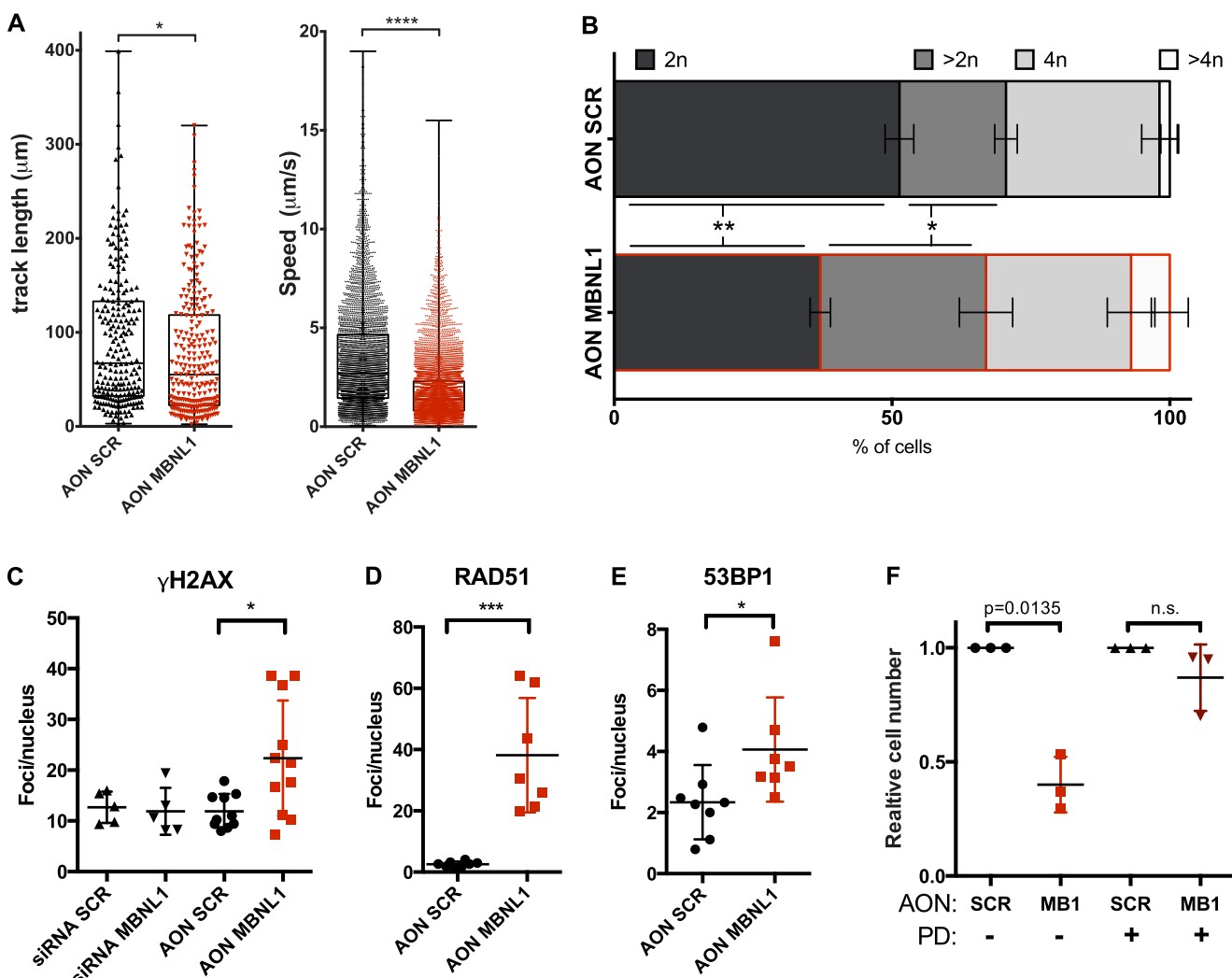

**Figure 6. Cells with MBNL1 Δex7 show reduced migration potential and increased markers of DNA damage.**
**(A)** PC3 migration changes in the collagen matrix over 14-h monitoring upon AON transfection. Left panel: track length of a single cell during the entire monitoring. Right panel: speed of a cell calculated comparing two consecutive frames. Data obtained for three different field of views/condition. Unpaired parametric *t* test. Data are representative of two independent experiments. **(B)** PI cell cycle profile of PC3 cells transfected with 100 nM AON at 48 h posttransfection. Statistics and quantification in Table S8. **(C)** γH2AX foci per nucleus in PC3 cells treated with MBNL1 siRNA (50 nM) and MBNL1 ex7 AON (100 nM) at 24 h from transfection. Every dot represents a quadrant with more than 15 cells present. Unpaired parametric *t* test. **(D)** RAD51 and (E) 53BP1 foci per nucleus in PC3 cells treated with MBNL1 siRNA (50 nM) and MBNL1 ex7 AON (100 nM) at 24 h from transfection. Every dot represents a quadrant with more than 15 cells present. Unpaired parametric *t* test. **(F)** PC3 cells arrested in G1 with CDK4/6 inhibitors show a reduction in cell death upon MBNL1 AON transfection at 48 h (75 nM). PD: PD03329691, CDK4/6 inhibitor (5 *μ*M). Parametric paired *t* test. Each dot represents an independent biological replicate.

medium for fluorescence with DAPI (H-1200; Vector Laboratories Inc.). Images were captured using a Lsm700 Zeiss laser scanning confocal microscope and analyzed with ImageJ software.

## Western blotting

Cells were lysed directly in 1× Lemmli buffer and sonicated with three short pulses before loading on SDS–polyacrylimide gels. Protein concentration was determined using the RC DC Protein Assay (Bio-Rad). 15–50 *μ*g of the protein extract was loaded on 6%–15% SDS–PAGE and subsequently transferred onto nitrocellulose membranes (10401396; Whatman Protran) using a semi-dry system. The membranes were

blocked either in 5% dry milk (70133; Sigma-Aldrich)/PBS or 5% BSA (0219989890; MP Biomedicals)/PBS and incubated overnight with primary antibodies, followed by secondary HRP-conjugated secondary antibodies (Table S9). The blots were developed using SuperSignal West Pico or Femto chemiluminescent substrate (34080; Thermo Fisher Scientific). Band intensities were quantified using ImageJ software (Schneider et al, 2012). Antibodies used are listed in Table S9.

## Immunoprecipitation

COS7 cells were grown in high-glucose DMEM (Lonza) supplemented with 10% FBS (Sigma-Aldrich) and 1× antibiotic/antimycotic

(Sigma-Aldrich). Cells were grown at 37 °C in a 5% atmosphere of $CO_2$ on 10 cm$^2$ plates and were co-transfected with Lipofectamine 3000 (according to the producer's protocol; Invitrogen) with 5 $\mu$g of GFP- and 5 $\mu$g of mCherry-fused particular MBNL1-expressing plasmid (except where a GFP-only vector was applied). Cells were harvested after 24 h and their lysates treated with RNase A were immunoprecipitated using GFP-Trap magnetic beads (2 h at 4°C; ChromoTek). All samples were mixed with loading buffer but not denatured at high temperature. PAGE filtration was conducted with 4%–12% gradient acrylamide gels (Invitrogen). Western blot was performed with either anti-mCherry monoclonal rat primary Ab (1:1,000, 1 h, RT; ChromoTek) or anti-GFP polyclonal rabbit primary Ab (1:1,000, 1 h, RT; Santa Cruz). mCherry-/GFP-fused MBNL1 isoforms migrate at ~70 kD (MBNL1 variant without ex7 weighs ~1 kD less).

## RNA-seq library preparation and splicing analysis

For RNA-seq library preparation, we followed the Illumina TruSeq RNA Sample Preparation Kit v2 manual. At least 70 million, 75-bp-long paired-end reads were mapped to the GRCh37/hg19 version of the human genome per replicate using STAR 2.4.2a (doi: 10.1093/bioinformatics/bts635) using the default parameters. Differential expression analysis was performed using Cuffdiff 2.2.1 (Trapnell et al, 2012) and only genes with a *P*-value and false discovery rate (FDR) of less than 0.05, an expression higher than 1 fragments per kilobase of transcript per million mapped reads, and a fold change of greater than 2 were labelled as significantly differentially expressed. To determine differential splicing events, rMATS 3.0.9 (Shen et al, 2014) was used counting junction reads and reads falling into the tested region within ENSEMBL v72 gene definitions. Splicing events were labelled significant if the *P*-value and FDR were lower than 0.05 and the minimum inclusion level difference as determined by rMATS was higher than 0.2. All other parameters were left at the default value. GO analysis was performed using Metascape (Tripathi et al, 2015) (http://metascape.org) keeping default parameters. For the differential expressed genes, all the significant events were considered, with a cutoff of 1.5-fold difference in expression between samples. For the differentially spliced genes, all the significant events were considered, applying a further cutoff of ±0.2 ΔPSI between the samples.

## Functional annotation

Functional annotation, GO, and graphical representations of the significant RNA-seq genes were performed with Metascape (Tripathi et al, 2015).

## Agarose gel–analyzed PCR

Primers were designed using CLC Main Workbench 5 (https://www.qiagenbioinformatics.com/products/clc-main-workbench/) and listed in Table S9. 2X DreamTaq Mastermix (EP1701; Thermo Fisher Scientific) was used as a PCR mix in a total volume of 25 $\mu$l. Primers (from IDT) were used at the final concentration of 1 $\mu$M. All PCRs were performed at an initial holding temperature of 95°C for 3 min, 26–32 cycles of denaturation at 95°C for 45 s, annealing at 58°C for 30 s and elongation at 72°C for 40 s, and a final elongation

temperature of 72°C for 4 min and 4°C holding temperature. PCR products were run in 1.7%–2.5% agarose gels, and 7 $\mu$l of ethidium bromide for every 100 ml of agarose solution was prepared. The PCR products were then loaded onto the gels and bands on gels were visualized using ImageQuant RT ECL imager (GE Healthcare). Band intensities were quantified using ImageJ software (Schneider et al, 2012).

## Real-time qRT–PCR and TaqMan PCR

Total RNA was isolated from the cells using the PureLink RNA Mini Kit (12183-018A; Ambion). 1 $\mu$g RNA was used to prepare cDNA using the Vilo cDNA kit (Invitrogen). The cDNA prepared was subjected to qRT–PCR (CFX 96 Touch; Bio-Rad), using SYBR Green PCR Supermix (Invitrogen) or TaqMan Universal PCR Master Mix (#4304437; Thermo Fisher Scientific). Data were expressed as relative mRNA levels normalized to housekeeper (GAPDH) expression levels in each sample. The primer and TaqMan probe sequences are in presented in Table S9.

## Fragment length analysis PCR (FLA-PCR)

2× DreamTaq Mastermix (EP1701; Thermo Fisher Scientific) was used as a PCR mix in a total volume of 25 $\mu$l. A specific 5'-FAM-labelled primer was used, as listed in Table S9. The final concentration of the FAM-labelled primer was 200 nM and the cDNA template added to the master mix has been kept the same for all the experiments (33 ng). The thermal cycler was set for an initial step of 94°C for 5 min, then 30 three-step cycles (94°C for 30 min, 56°C for 45 min, and 72°C for 45 min), and subsequent eight three-step cycles (94°C for 30 min, 52°C for 45 min, and 72°C for 45 min) with a final cycle at 72°C for 3 min. 1 $\mu$l of the PCR product was added to 9.1 $\mu$l of HiDi + Liz markers premix (GeneScan 500/1,200 LIZ dye Size Standard, Thermo Fisher Scientific #4322682/4379950, optimized to give 1K RFU peak height) and then to each sample. The mix was denatured at 96°C for 3 min, cooled at 4°C for 3 min, and run in the Thermo Fisher Scientific 3730XL DNA analyzer. GeneMapper 5 (Thermo Fisher Scientific) software was used to analyze the peaks.

## siRNA screening

Dharmacon On-Target plus siRNA pools against the genes listed in Fig 2A were purchased in the 96-well format and resuspended at 500 nM concentration in 96-well plates treated for tissue culture. For each well was prepared a mix of 25 $\mu$l of Optimem (#31985062; Thermo Fisher Scientific) and 0.3 $\mu$l of Lipofectamine RNAiMAX Transfection Reagent (#13778030; Thermo Fisher Scientific) following an incubation of 5 min at room temperature. In the meanwhile, 25 $\mu$l of Optimem was added directly into every well of the 96-well plate containing the siRNAs. After incubation, 25 $\mu$l of the Optimem–RNAiMAX mix was added to every well of the plate, following an incubation at RT for 20 min on a gentle shaker (300 rpm). PC3 cells were trypsinized, washed in PBS, and resuspended in Optimem at the final concentration of 18,000 cells/50 $\mu$l media. 50 $\mu$l of the cell suspension was then added to the siRNA–RNAiMAX 96 plate, pipetting gently for two times, and then placed in a cell incubator overnight. The medium was replaced in

the morning with normal RPMI + 10% FBS. 48 h after the transfection, cells were lysed and RNA was extracted and retrotranscribed using the CellsDirect One-Step qRT–PCR kit (#11753100; Thermo Fisher Scientific) following the manufacturer's instructions. The RT/TaqMan PCR to detect MBNL1 isoforms containing or excluding ex7 together with the housekeeper GAPDH was performed with 7.5 $\mu$l of cDNA obtained from the CellsDirect kit. The TaqMan mix was made by mixing 10 $\mu$l of Mastermix, 0.5 $\mu$l of SuperScriptIII/Platinum, 1 $\mu$l of primer mix (Table S9, from 10 $\mu$M stock), 0.5 $\mu$l of TaqMan probes mix (Table S9, from 5 $\mu$M stock), and 0.5 $\mu$l of water per sample. Cycles were set in the following way on the BioRad CFX96 Touch: 50°C for 18 min, 95°C for 2 min, and then a two-step cycle at 95°C for 5 min and 60°C for 30 min to be repeated for 40 times.

### Cell viability and counting

Cell counting was performed after trypsinization of the cells and the addition of 0.4% trypan blue (1:1, #15250061; Gibco) using a Countess II FL automated cell counter (Thermo Fisher Scientific), accounting only for the live cells. Cell viability was assessed using the CellTiter 96 Aqueous One solution assay (G3582; Promega) following the manufacturer's instructions. Reading of the absorbance was performed after 30 min of incubation at 37°C with a Tecan Infinite microplate reader.

### 3D hydrogel migration assay

A commercially available microfluidic device (AIM Biotech; DAX-1) was used to culture genetically modified prostate cancer cell lines (PC3 cells transduced with pBOBI-H2B-GFP, a kind gift from Philipp Kaldis Lab). Collagen hydrogel was prepared for injection in the dedicated microfluidic region with the following master composition: 2.5 mg/ml type-I collagen gel solution containing 20 $\mu$l of the homogeneously dissociated prostate cell line at 4 × 10$^6$ cells/ml (stained with cell tracker Orange CMRA; Thermo Fisher Scientific), mixed with 20 $\mu$l of 10× PBS, 4 $\mu$l of NaOH (0.5 N), 129.2 $\mu$l of collagen type I (Corning), and 22.9 $\mu$l of cell culture water. 10 $\mu$l of hydrogel master aliquot was used for injection in the dedicated device section; the gel was then thermally cross-linked at 37°C for 40 min in the cell incubator. After gel polymerization, media channels were filled with RPMI + 10% FBS. The devices were then placed on the microscope stage incubator (Tokai Instruments) at 5% $CO_2$ and 37°C to start the live-imaging (time-lapse) confocal acquisition (LSM7800 confocal microscope; Zeiss). The microscope was programmed to acquire Z-stacks (250 $\mu$m) of the selected regions at the stated time intervals (15 min). All images acquired from the confocal microscope systems were visualized and analyzed using Imaris (Bitplane).

## Data Access

Gene Expression Omnibus accession number: GSE114383.

## Supplementary Information

## Acknowledgments

We thank Prof. Philipp Kaldis, Dr. Xavier Bisteau, Prof. Uttam Surana, and Dr. Anand Jeyasekharan for sharing protocols and reagents and for their helpful discussions. We thank Wolfson Centre for Inherited Neuromuscular Disease for donating the MBNL1 antibodies, Biological Resource Centre (BRC) for sharing facilities, and Ah Keng Chew for technical support. We are grateful to the Genome Institute of Singapore–A*STAR (GIS) Genome Sequencing Team for their help with high-throughput sequencing and to the entire Guccione's laboratory for critical discussion. This work was supported by A*STAR Graduate Award–Singapore INternational Graduate Award (AGA-SINGA) fellowship to T Tabaglio and PA Goy. E Guccione acknowledges support from NRF-CRP17-2017-06.

### Author Contributions

T Tabaglio: conceptualization, data curation, formal analysis, validation, investigation, visualization, project administration, and writing—original draft, review, and editing.
DHP Low: data curation, formal analysis, and investigation.
WKL Teo: investigation.
PA Goy: data curation and investigation.
P Cywoniuk: investigation.
H Wollmann: investigation.
J Ho: investigation.
D Tan: investigation.
J Aw: investigation.
A Pavesi: formal analysis.
K Sobczak: investigation.
DKB Wee: data curation.
E Guccione: conceptualization, resources, supervision, funding acquisition, project administration, and writing—original draft, review, and editing.

### Conflict of Interest Statement

The authors declare that they have no conflict of interest.

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
