## [Reviewer comments · Life Science Alliance]

MBNL1 alternative splicing isoforms play opposing roles in cancer

Tommaso Tabaglio, Diana HP Low, Winnie Koon Lay Teo, Pierre Alexis Goy, Piotr Cywoniuk, Heike Wollmann, Jessica Ho, Damien Tan, Joey Aw, Andrea Pavesi, Krzysztof Sobczak, Dave Keng Boon Wee and Ernesto Guccione
DOI: 10.26508/lsa.201800157

Review timeline:

Submission Date:	14 August 2018
Revision Received:	14 August 2018
Editorial Decision:	17 August 2018
Accepted:	29 August 2018

Report:

(Note: Letters and reports are not edited. The original formatting of letters and referee reports may not be reflected in this compilation.)

Please note that the manuscript was previously reviewed at another journal and the reports were taken into account in the decision-making process at Life Science Alliance. Since the original reviews are not subject to Life Science Alliance's transparent review process policy, the reports and author response cannot be published.

1st Editorial Decision

17 August 2018

Thank you for submitting your revised manuscript entitled "MBNL1 alternative splicing isoforms play opposing roles in cancer". Your manuscript was previously reviewed at another journal, and the editors shared your manuscript and the reviewer reports with us with your permission. Based on these reports we had invited you to submit a revised version of your work for publication in Life Science Alliance.

The reviewers who evaluated your work at the other journal had noted that your work is robust pending several clarifications, but that quantifications are missing and that the insight provided into the role of MBNL1 splice variants in cancer, in cell death regulation and in DNA damage is limited. You provided a point-by-point response to us and you revised your manuscript to include quantifications and to clarify several issues the reviewers had noted. We assessed your revised work and appreciate the introduced changes. We would be thus happy to publish your paper in Life Science Alliance pending final revisions necessary to meet our guidelines:

- please add a callout for SFig3K
- please provide the Supplementary Tables 1-9
- please upload all figures as individual files of high resolution
- please fill in all questions in our manuscript submission system and make sure that all affiliations and author contributions are provided for all authors (please see also <http://www.icmje.org/recommendations/browse/roles-and-responsibilities/defining-the-role-of-authors-and-contributors.html> for the authorship definition that our journal adheres to)
- please provide an ORCID id, you should have received an email with instructions on how to do so.

Thank you for submitting your Research Article entitled "MBNL1 alternative splicing isoforms play opposing roles in cancer". It is a pleasure to let you know that your manuscript is now accepted for publication in Life Science Alliance.

Congratulations on this interesting work.